# A Controlled Clinical Study of Intensive Neurorehabilitation in Post-Surgical Dogs with Severe Acute Intervertebral Disc Extrusion

**DOI:** 10.3390/ani11113034

**Published:** 2021-10-22

**Authors:** Ângela Martins, Débora Gouveia, Ana Cardoso, Carla Carvalho, Tiago Coelho, Cátia Silva, Inês Viegas, Óscar Gamboa, António Ferreira

**Affiliations:** 1Faculty of Veterinary Medicine, Lusófona University, Campo Grande, 1300-477 Lisboa, Portugal; 2Animal Rehabilitation Center, Arrábida Veterinary Hospital, Azeitão, 2925-583 Setúbal, Portugal; deborahisabel@msn.com (D.G.); anacardosocatarina@gmail.com (A.C.); mv.carla.c@gmail.com (C.C.); tiagoccoelho@netcabo.pt (T.C.); catiamsilva@outlook.pt (C.S.); inesviegas@gmail.com (I.V.); 3CIISA—Centro Interdisciplinar-Investigação em Saúde Animal, Faculdade de Medicina Veterinária, Av. Universidade Técnica de Lisboa, 1300-477 Lisboa, Portugal; aferreira@fmv.ulisboa.pt; 4Superior School of Health, Protection and Animal Welfare, Polytechnic Institute of Lusophony, Campo Grande, 1300-477 Lisboa, Portugal; 5Faculty of Veterinary Medicine, University of Lisbon, 1300-477 Lisboa, Portugal; ogamboa@fmv.ulisboa.pt

**Keywords:** spinal cord injury, locomotor training, functional electrical stimulation, transcutaneous electrical SC stimulation, 4-aminopyridine, neurorehabilitation, acute dogs

## Abstract

**Simple Summary:**

This study explores the potential intensive neurorehabilitation plasticity effects in post-surgical paraplegic dogs with severe acute intervertebral disc extrusion aiming to achieve ambulatory status. The intensive neurorehabilitation protocol translated in 99.4% (167/168) of recovery in deep pain perception-positive dogs and 58.5% (55/94) in deep pain perception-negative dogs. There was 37.3% (22/59) spinal reflex locomotion, obtained within a maximum period of 3 months. Thus, intensive neurorehabilitation may be a useful approach for this population of dogs, avoiding future euthanasia and promoting an estimated time window of 3 months to recover.

**Abstract:**

This retrospective controlled clinical study aimed to verify if intensive neurorehabilitation (INR) could improve ambulation faster than spontaneous recovery or conventional physiotherapy and provide a possible therapeutic approach in post-surgical paraplegic deep pain perception-positive (DPP^+^) (with absent/decreased flexor reflex) and DPP-negative (DDP^−^) dogs, with acute intervertebral disc extrusion. A large cohort of T10-L3 Spinal Cord Injury (SCI) dogs (*n* = 367) were divided into a study group (SG) (*n* = 262) and a control group (CG) (*n* = 105). The SG was based on prospective clinical cases, and the CG was created by retrospective medical records. All SG dogs performed an INR protocol by the hospitalization regime based on locomotor training, electrical stimulation, and, for DPP^−^, a combination with pharmacological management. All were monitored throughout the process, and measuring the outcome for DPP^+^ was performed by OFS and, for the DPP^−^, by the new Functional Neurorehabilitation Scale (FNRS-DPP^−^). In the SG, DPP^+^ dogs had an ambulation rate of 99.4% (*n* = 167) and, in DPP^−^, of 58.5% (*n* = 55). Moreover, in DPP^+^, there was a strong statistically significant difference between groups regarding ambulation (*p* < 0.001). The same significant difference was verified in the DPP^–^ dogs (*p* = 0.007). Furthermore, a tendency toward a significant statistical difference (*p* = 0.058) regarding DPP recovery was demonstrated between groups. Of the 59 dogs that did not recover DPP, 22 dogs achieved spinal reflex locomotion (SRL), 37.2% within a maximum of 3 months. The progressive myelomalacia cases were 14.9% (14/94). Therefore, although it is difficult to assess the contribution of INR for recovery, the results suggested that ambulation success may be improved, mainly regarding time.

## 1. Introduction

Spinal cord injury (SCI) leads to temporary or permanent changes in the motor, sensory, and automatic functions [1]. Following SCI, it is possible to activate and strengthen connections until spontaneous recovery [2], although it is not clear how intensive neurorehabilitation (INR) training can induce plasticity.

INR is a field of physical medicine and rehabilitation that is based on the evidence of signal transmission throughout the lesion caudally and rostrally, which can be detected by electromyography [3,4,5]. Thus, the main goal of INR is to trigger the central axon pathways, traversing the lesion, by synaptic stimulation [4,6] and to stimulate propriospinal connections that may bypass the injury site and possibly mediate recovery [6,7].

The INR intends to facilitate central nervous system (CNS) reorganization at multiple levels. Mostly, it intends to achieve a balance between the central pattern generators (CPG), spinal rhythm-generating circuitry plasticity, persistent descending pathways [8,9], and the sensory feedback, which stimulates the CPG [10,11,12,13]. While descending pathways can produce start–stop signals, which are essential for coordination and posture [12], propriospinal interneurons may promote new intraspinal circuits after severe SCI [2]. Therefore, INR, in addition to spontaneous recovery, could induce recovery based on similar mechanisms.

The INR protocols are based on a multimodal approach, which includes locomotor training [10,14,15,16], electrostimulation protocols [2,17,18,19,20,21,22], and, in circumstances of absent deep pain perception (DPP^_^), pharmacological management [23,24,25,26,27,28,29].

Locomotor training promotes the activation of the spinal locomotor circuitry, which interacts dynamically with afferent inputs from the receptors of muscles, joints, and skin [16,30,31].

Functional electrical stimulation (FES) promotes an unnatural recruitment pattern of muscle fibers, starting with large-diameter ones (fast neurons) instead of fatigue-resistant slow motoneurons. In addition, FES may stimulate new intraspinal circuits and persistent residual spinal circuits [3,32,33,34,35].

Furthermore, transcutaneous electrical stimulation can promote neuromodulation effects in the spinal cord locomotor network of the lumbar region [36,37,38] and plays a role in the spinal rhythm-generating circuitry [39,40].

The use of multidisciplinary approaches in human patients and in animal models may improve ambulation [41,42]. A similar approach could be applied in DPP^−^ and DPP^+^ dogs in a clinical setting.

The main aim of this study was to verify if INR could improve the ambulatory status faster than spontaneous recovery or conventional physiotherapy with a minimum sensory deficit and, also, to provide a possible therapeutic approach in post-surgical dogs with acute intervertebral disc extrusion (IVDE).

This study hypothesized that a multimodal training protocol may improve ambulation in paraplegic dogs DPP^−^ and DPP^+^.

## 2. Materials and Methods

This study was conducted between May 2011 and May 2020 at the Arrábida Veterinary Hospital (Arrábida Animal Rehabilitation Center, Setúbal, Portugal), after approval by the Lisbon Veterinary Medicine Faculty Ethics Committee and after the owners’ consent.

### 2.1. Participants

This was a retrospective controlled clinical study using a large cohort of dogs (*n* = 367). The study group (SG) was composed of 262 dogs, and the control group (CG) consisted of 105 dogs.

Dogs from the SG were prospective clinical cases that were monitored, and all data was collected and registered during the rehabilitation process.

The 105 dogs of the CG were selected from the Arrábida Veterinary Hospital data management system, and the medical records were retrospectively investigated. These medical records focused on breed, age, weight, etiology, surgical treatment, and a neurological examination at admission and at medical discharge.

All the dogs (*n* = 367) had compressive myelopathy by extruded material (Hansen Type I IVDE) with T10-L3 segment injury diagnosed by computed tomography (CT) with/without myelogram or magnetic resonance imaging (MRI). The standard CT signs observed were hyperattenuating material inside the vertebral canal, a loss of epidural fat, and deformity of the spinal cord. Dogs were included with two different imaging patterns: acute extruded mineralized nucleus pulposus or acute extrusion of nucleus pulposus with hemorrhage. The myelogram findings reported lateralized focal spinal cord compression or herniation into the ventral subdural space. Regarding MRI, only some dogs of both groups needed this type of exam, showing hyperintensity on T2W images, which could correlate with the severity and presentation of the clinical neurological findings, such as necrosis, myelomalacia, inflammation, edema, and intramedullary hemorrhage.

Dogs were treated by hemilaminectomy 3–5 days after injury (first clinical sign appearance) and classified with the Frankel Modified Scale (FMS) as grade 0 (DPP^−^) or 1 (DPP^+^) before and after surgery.

Regarding the SG, all dogs were admitted to the rehabilitation center less than 7 days after surgery. One hundred and sixty-eight dogs were paraplegic DPP^+^ with an absent/decreased flexor peripheral reflex and classified as grade 1 according to the Open Field Score (OFS) [43].

The other 94 dogs were paraplegic DPP^−^, evaluated according to a new score scale—the Functional Neurorehabilitation Scale for dogs with Thoracolumbar SCI without Deep Pain Perception (FNRS-DPP^−^) [44]—and classified as grade 0 or 1 (Figure 1). The dogs were classified with FNRS-DPP^−^ grade 1 only if the patellar reflex was present.

Concerning the CG, 62 dogs were paraplegic DPP^+^ with an absent/decreased flexor peripheral reflex and classified with OFS 1, and 43 dogs were paraplegic DPP^−^.

This group was subjected to cage rest, passive range of motion exercises (PROMS), massages, supported postural standing, gait stimulation, and neuromuscular electrical stimulation (NMES).

All animals in the study were less than 7 years old and weighed less than 25 kg. Most were of chondrodystrophic breeds and lacked other concomitant diseases. Dogs were excluded if they presented other SCI, were outside T10–L3, or if they had a surgical approach before 3 days or more than 5 days after the injury. Moreover, animals were excluded if they had higher OFS scores (>1) or higher grades of FNRS-DPP^−^ (>1). Regarding the SG, dogs that were admitted to the rehabilitation center more than 7 days post-surgery were also excluded.

### 2.2. Study Design

The 367 dogs were subjected to a neurorehabilitation consultation and evaluated according to their history and physical and neurorehabilitation examinations. In the neurorehabilitation examination, the following were examined: mental status, posture, postural reactions, spinal reflexes, cutaneous trunci muscle reflex, spinal palpation, pain perception, and gait. For the SG (*n* = 262), the gait for DPP^+^ dogs was evaluated using the OFS and for DPP^–^ dogs using the FNRS-DPP^−^.

Regarding gait, SG dogs were evaluated during the same time (3 to 7 p.m.) on the same 4-m ceramic surface floor by the same observer always in the same position. The neurorehabilitation examination was performed in a controlled environment, without external noise and with a restricted number of people. Study participants underwent an accurate evaluation regarding DPP, tested on the medial and lateral digits of the hindlimb bilaterally and on the tip and base of the tail. Superficial sensitivity was assessed on the S1 and S2 dermatomes with 12-cm Halsted mosquito forceps. The perineal region, including the bulbocavernosus reflex, was also assessed.

INR protocol started 24 h after admission for all dogs. Within this period, the dogs were prepared for training and rehabilitation modalities (e.g., trichotomy), adapted to the treadmill, and presented to the rehabilitation team, who attended to their tender, love, and care needs.

All SG dogs were subjected to weekly evaluations by a certified canine rehabilitation professional (CCRP) examiner/instructor at the University of Tennessee. The data were recorded (Canon EOS Rebel T6 1300 D camera), and all images were then revised by another CCRP instructor and a non-CCRP neurologist at the Lisbon Veterinary Medicine Faculty. SG dogs were admitted to an INR through a hospitalization regimen for a maximum period of 3 months, as described in Figure 2.

As acute IVDE dogs, all SG and CG dogs were under a nonsteroidal anti-inflammatory treatment with carprofen (2.2 mg/kg BID) or meloxicam (0.1mg/kg SID) for 5 days or corticosteroids with prednisolone (0.5 mg/kg per os BID/SID) for 3–5 days. After 7 days post-surgery, this treatment was totally discontinued in both the SG and CG. The SG dogs remained only with the INR protocol, and the CG remained with cage rest, PROMS, massage, postural standing, gait stimulation, and NMES.

### 2.3. Interventions

Dogs started INR training with an association of locomotor training and electrical stimulation 24 h after admission. Pharmacological management was only applied in DPP^−^ dogs that did not recover DPP until day 30 but demonstrated a flexion–extension locomotor pattern.

#### 2.3.1. Locomotor Training

Dogs were accustomed to the land treadmill and started with a higher body weight support (BWS) (60–80% body weight) [45], which was decreased with the load tolerance [46], always supervised by a rehabilitator.

BWS was achieved with a harness, allowing quadrupedal step training as part of their daily protocol. However, when some resistance was offered, a change to bipedal step training was required [47,48]. During bipedal training, the forelimbs rested on a platform raised above the treadmill belt [49] while the perineal area was stimulated by suspending and crimping the tail [50] or with assisted bicycle hindlimb movements [51].

For each training session, variables such as the walking speed and duration were increased and recorded, starting from 0.8 km/h (0.5 mph) to a maximum of 1.9 km/h (1.2 mph) [52,53,54] over 5 min (4–6 times/day, 6 days/week), with the aim of reaching 20 min (2 times/day, 6 days/week) [55].

The clinical study participants in quadruped training received similar stimulation with the same speed and frequency variables. The goal in this group was to reach 30–40 min (2 to 3 times/day, 6 days/week) [56]. The treadmill slope was then elevated from 10° [47] to 25° [57] to encourage forelimb–hindlimb coordination [58].

All patients began underwater treadmill training 2–7 days after admission. All started with a water temperature of 26 °C [59,60] and a 5-min walk until reaching 1 h of training once a day (5 days/week) from 1–3.5 km/h (2.2 mph) [61,62] while overtraining signs were monitored.

Furthermore, kinesiotherapy exercises were performed, such as: Cavaletti rails (3 rounds 3 times/day), balance boards (1 to 2 min and increase to 5 min 3 times/day—with or without BWS), and gait stimulation on different surfaces (1–5 min 3 times/day).

#### 2.3.2. Electrical Stimulation

Electrical stimulation protocols were used to manage the pain through interferential electrical stimulation (IES): increase muscle strength and neural connections with FES and increase the potential descending pathway depolarization with transcutaneous electrical spinal cord stimulation (TESCS).

Interferential electrical stimulation

This technique is a form of stimulation that has two separate channels and uses alternating currents [63] through four electrodes placed on the skin near the region of spinal hyperesthesia and crossed at a 90° angle with the following parameters: acute pain, 80–150 Hz, and 2–50 ms; chronic pain, 1–10 Hz, 100–400 ms [64,65], once a day (Figure 3).

Functional electrical stimulation

This neuromodulation modality uses a short electrical pulse sequence, resulting in spinal reflexes. It aims to stimulate the lower motoneuron near the motor region or through peripheral afferent stimulation [66,67,68]. This modality was performed in all patients with superficial electrodes, using a segmental technique. One electrode was placed on the skin region corresponding to L7–S1, and the other electrode was placed near the ventromedial motor region of the hindlimb flexor muscle group using a pulsated and biphasic current.

The parameters were 60 Hz, 6–24 mA [69,70]; 1:4 duty cycle; and 2–4-s ramp up, 8-s plateau, and 1 to 2-s ramp down [64] over 10 min. This routine was performed 2 to 3 times/day (5 days/week) and was discontinued according to each patient’s neurological improvement. After the dogs were subjected to the FES modality, they had a therapeutic window of 40 min, during which the locomotor training was conducted [71,72,73].

Transcutaneous electrical spinal cord stimulation

All the patients underwent TESCS 3 times/day (5 days/week), which was gradually discontinued when the flexion–extension locomotor pattern appeared. The surface electrodes were placed on the paravertebral muscles (one electrode at T11to T12 and the other at L7–S1) [74,75,76,77] with a continuous current of 50 Hz, 2 mA for 10 min [36,37,78,79,80] (Figure 4).

#### 2.3.3. Pharmacological Management

During the 3rd to 4th weeks (T4–T5) of the training, if the flexion–extension locomotor pattern was present with a DPP^−^ result (tested on the medial and lateral digits of the hindlimb bilaterally and on the tip and base of the tail), it was added with the owner’s consent and is called pharmacological management.

4-aminopyridine (4-AP) was administered, a K+ channel-blocking compound [25,26,29,81,82,83,84], under the following regime: 0.3 mg/kg per os BID for 3 days, 0.5 mg/kg BID for 3 days, 0.7 mg/kg BID for 3 days, and 1.1 mg/kg BID for 21 days. The 4-AP protocol was implemented for a maximum of 2 months.

If any side effects (seizures, diarrhea, and vomiting) occurred, the dogs were immediately treated and withdrawn from the clinical study.

INR protocol application was consistently performed within the patient’s cardiorespiratory capacity and, according to the evolution observed during the functional neurorehabilitation examination and OFS (DPP^+^ dogs) or FNRS-DPP^−^ (DPP^−^ dogs) assessment over a maximum of 3 months.

#### 2.3.4. Supportive Measures

Most of the dogs in the clinical study had neurogenic bladders. Thus, bladder expression was performed 3 to 4 times/day [85], and the urine was monitored daily for odor and color changes. If there was a suspected urinary tract infection, urine culture (by cystocentesis) and specific antibiotic treatment were administered.

The dogs were maintained under a full-time hospitalization regime. They were able to rest on soft beds with multiple disposable absorbent pads and were encouraged to maintain sternal recumbency. Dogs were fed three times per day with an intake increase of 30% and oral hydric support of 100–120 mL/kg; after, the resistance training was alternated with strength training, according to the patient’s needs. At the end of the day, class IV laser therapy was administered to reduce pain at the trigger points [86].

All dogs started training at 9:00 a.m. and finishing at 7:00 p.m. They were assisted only by veterinarians and veterinary nurses who had taken the CCRP course.

### 2.4. Outcome Measures

SG dogs were assessed by neurological examination every 5–7 days by the same certified CCRP examiner/instructor. The measured outcomes, including the OFS or FNRS-DPP^−^, were evaluated at different time points: admission (T1), day 3 (T2), day 7 (T3), day 15 (T4), day 30 (T5), day 45 (T6), day 60 (T7), day 75 (T8), and day 90 (T9) after starting the INR. Follow-ups were performed after 8–10 days (F1), 1 month (F2), 6 months (F3), and one year (F4) (Figure 2).

The presence of DPP, the flexor reflex, flexion–extension locomotor pattern, and postural standing ability were monitored in these evaluations, which facilitated the establishment of an accurate and systematic evaluation of ambulation recovery among dogs.

Ambulation was defined as the patient’s ability to stand up, maintain postural standing, take at least ten steps without assistance or weight support on any walking surface, and obtain voluntary or automatic micturition and defecation.

Autonomous ability in movement control suggests that parts of the brain and, also, the spinal cord may presumably activate movements with some conscious control [87].

“Spinal Reflex” Locomotion (SRL) can promote the autonomous ability to stand up and walk in “DPP^−^ dogs”, maintaining some coordination between the forelimbs and hindlimbs, perhaps by propriospinal system reorganization, promoting the dog´s ability to not fall when changing directions on a non-slippery floor.

DPP^−^ SG dogs were considered ambulatory if they showed an SRL score of ≥14 on FNRS-DPP^−^. For DPP^+^ SG dogs, ambulation was considered when the OFS was ≥11.

At the end of the study, dogs that become DPP^+^ or DPP^−^ but with functional SRL or non-functional SRL were discharged and released into the owner´s guardianship.

DPP^−^ dogs that showed signs compatible with progressive myelomalacia (PM), and upon the owner’s request, were euthanized by induction with Propofol IV after they fell asleep, followed by Pentobarbital IV, within a quiet room.

CG dogs were evaluated at admission and at discharge, regarding the neurological examination and FMS. For the CG, ambulation was considered within the same parameters as described for the SG.

### 2.5. Statistical Analysis

Database and statistical analyses were performed through Microsoft Office Excel 2016 software (Microsoft Corporation, Redmond, WA, USA). Quantitative, qualitative, and categorical data were analyzed using IBM SPSS Statistics software, version 22 (International Business Machines Corporation, Armonk, NY, USA), and the results were interpreted at the *p* ≤ 0.05 level of significance. The categorical data were presented as frequencies and proportions (95% confidence interval).

Regarding the data, normality tests and histograms were performed, showing a normal distribution.

Chi-square tests were performed to demonstrate the presence of statistically significant differences between the SG and the CG. Independent *t*-tests of the samples were also performed to compare groups regarding population characterization.

In the SG, the estimated marginal means and interaction plots for comparison at each time point regarding the OFS scores or FNRS-DPP^–^ scores were performed.

## 3. Results

The total sample of this study was 367 participants, distributed into an SG (*n* = 262) and a CG (*n* = 105). Acknowledging that the size of the SG and the CG could impair the analysis, Cohen’s d tests were performed, when applicable, and the effect size was always small.

In each group, dogs were admitted in both grade 0 (DPP^−^) and 1 (DPP^+^), according to the FMS. Thus, the results are divided as follows.

### 3.1. DPP^+^ Dogs

Grade 1 (DPP^+^) was represented by 62.7% of the total population (*n* = 230). Of these, 168 dogs belonging to the SG (73%) and 62 (27%) were controls.

Regarding population characterization, the most common breed among the DPP^+^ dogs was the Mixed breed (18.7%; *n* = 43), followed by the French Bulldog (17.4%; *n* = 40) and the Dachshund 11.3%; *n* = 26). However, 68.3% (157/230) were of a chondrodystrophic breed.

In the total DPP^+^ population (SG + CG), there were 59.6% males and 40.4% females, and, regarding the neuro-location, the main region was T12 to T13 (30.4%), followed by T13–L1 (26.1%). The mean age of the population was 4.07 years (median of 4.00), and the mean weight was 8.79 kg (median of 8.00). Individual means and medians of the age and weight for both groups are represented in Table 1.

Concerning ambulation, 99.4% (*n* = 167) of the dogs became ambulatory in the SG, whereas only 75.8% (*n* = 47) achieved ambulation in the CG. Furthermore, while only one dog did not achieve ambulation in the SG, 24.2% (*n* = 15) were considered non-ambulatory in the CG. Comparing both groups, there was a strong difference regarding statistical significance (*X*^2^ (1, *n* = 230) = 38.963, *p* < 0.001).

Dogs of the SG had clinical discharge as follows: 23.2% (*n* = 39) in T3, 44.6% (*n* = 75) in T4, 25% (*n* = 42) in T5, 4.8 % (*n* = 8) in T6, 0.6% (*n* = 1) in T7, and 1.8% (*n* = 3) in T8 (Figure 5 and Figure 6).

Moreover, in each time point and follow-up consultations, the OFS values of each dog in the SG were registered, and their estimated mean values revealed a progressive evolution in time (Figure 7), achieving a maximum value in time point day 30 with an OFS mean of 11.8 (*n* = 54). The graph obtained showed a slight decrease in time point day 45 with an OFS mean of 10.7 (*n* = 12).

### 3.2. DPP^−^ Dogs

Grade 0 (DPP^−^) was represented by 37.3% of the total population (*n* = 137), with 94 dogs belonging to the SG and 43 dogs to the CG.

From these 137 dogs, the French Bulldog was the most common at 27% (*n* = 37), followed by the Mixed breed at 25.5% (*n* = 35) and the Dachshund at 15.3% (*n* = 20). Most dogs were chondrodystrophic, representing 72.3% of the DPP^−^ population (*n* = 99), with 39.4% females and 60.6% males.

For the total DPP^−^ population (SG + CG), regarding the neuro-location, T12 to T13 represented 27%, followed by T13–L1 (19.7%) and L1–L2 (18.2%). Thus, most lesions occurred caudally to T12. The mean age of the population was 4.03 years (median of 4.00), and the mean weight was 8.14 kg (median of 8) (Table 2).

Having tested the equality of variances (nonsignificant), the *t*-test for independent samples in the categories age and weight revealed no significant difference between the two groups, SG and CG (*p* = 0.864 and *p* = 0.112), thus making them comparable. The means and medians of age and weight for both groups are represented in Table 2.

The DPP recovery in the SG was observed in 33.2% (35/94) and 21% (9/43) in the CG. Participants that did not recover their DPP were 62.8% (59/94) in the SG and 79% (34/43) in the CG. Thus, there was a tendency towards a significant statistical difference between groups (*X^2^* (1, *n* = 137) = 3.597; *p* = 0.058).

Of the 59 dogs that did not recover their DPP, 22 dogs regained ambulation through SRL, 37.3% (22/59) of them within a maximum of 3 months.

In regard to ambulation, 58.5% (*n* = 55) recovered their ambulatory status in the SG, while there were only 32.6% (*n* = 14) in the CG. On the other hand, 41.5% (*n* = 39) did not achieve ambulation in the SG and 67.4% (*n* = 29) in the CG. Therefore, the comparison between groups showed a strong difference with statistical significance regarding ambulation recovery (*X^2^* (1, *n* = 137) = 7.311; *p* = 0.007).

As mentioned earlier, from the total DPP^−^ population, 35 dogs recovered DPP: two dogs in T4, although they were withdrawn from the study by the owner’s decision; seven dogs in T5, achieving ambulation with grade 5 (FMS); five dogs in T7 (grade 5 FMS); and 21 dogs in T9 (grade 5 FMS) (Figure 8).

Regarding the 39 dogs that did not achieve ambulation, two left the study in T4, as mentioned above, and 37 were discharged in grade 0 (FMS). From these, in T3 (*n* = 3), T4 (*n* = 9), and T5 (*n* = 2), the dogs presented signs of PM. Thus, 14.9% (14/94) of the participants in this study presented with myelomalacia. Furthermore, in T7 and T9, the remaining 23 dogs were discharged in grade 0 (FMS).

For the dogs in the SG, after 30 days of INR, it was associated with pharmacological management with 4-AP. Thus, in the group of dogs that had DPP recovery and regained ambulation, 26 of them were after 4-AP. SRL was also obtained in T7 (*n* = 10) and T9 (*n* = 12), always after 4-AP administration.

Comparing each time point and follow-up, the dogs that did not recover DPP (*n* = 59) were monitored with the FNRS-DPP^–^ scale. The estimated mean values were recorded and are shown in Figure 9, demonstrating an ascending curve throughout the 90 days of INR, with the first maximum value achieved at the time point day 60. Moreover, in the 6-month follow-up, the mean score value was 11.3, which increased to 17.3 in the one-year follow-up (*n* = 11).

The 35 dogs that recovered their DPP were monitored according to the OFS, and the estimated means were registered (Figure 10), revealing an OFS of 0 until time point day 7, ascending from that day on with a maximum value at time point day 90 (OFS mean 12.3) and achieving a mean score of 13 in the 6-month follow-up that was maintained until one year.

## 4. Discussion

This retrospective-controlled clinical study intended to evaluate the effect of INR in the SG versus the CG in both DPP^+^ and DPP^−^ dogs.

The total population was 367 dogs, allowing an approximate level of power (1 − β) of 0.90 and an α (type I error) of 0.01. For the SG (*n* = 262), an approximate level of power (1 − β) of 0.90 and an α (type I error) of 0.05 [88] was possible.

The population sample of both the SG and CG was not easy to obtain, needing a total of nine years, given the strict and specific inclusion criteria. A pilot study was previously carried out with 84 dogs subjected to similar INR protocol guidelines as this study. The results were accepted for publication in the ISCOS meeting and were continued in this clinical study [89].

In acute IVDE dogs, it is challenging to differ recovery due to spontaneous plasticity or by possible induced plasticity due to rehabilitation techniques. Thus, this study explored the possible INR plasticity effects, considering that spontaneous neurological recovery usually reaches a plateau within the first few weeks after surgery [90,91,92].

Furthermore, selecting the participants—DPP^−^ dogs or DPP^+^, but with absent/decreased flexor peripheral reflex, similar to Jeffery et al. (2020) [93]—makes recovery harder to achieve. Participants were classified at admission using the FNRS-DPP^−^ scale with grades 0/1 or OFS 1, respectively, in order to fully understand the possible effects of INR.

The absence/decreased flexor reflex may be attributed to the incidence of spinal shock, which can lead to a secondary lesion within the spinal cord segments containing the hind limbs reflex circuitry; when this clinical sign persists over time, it may be consistent with descending myelomalacia.

At admission, FNRS-DPP^−^ grade 0 is attributed to the absence of any peripheral reflex in the hindlimbs, whereas dogs in grade 1 have a positive patellar reflex, making the possibility of PM less probable. In the future, it would be of interest to relate the presence of this reflex with the ability to achieve SRL.

All the DPP^−^ dogs underwent a strict evaluation regarding DPP—decreasing the variability and increasing the degree of accuracy in testing—which is essential for predicting outcomes [94,95,96].

The SG and CG could be considered analogous and a homogenous sample population, making comparison possible. Both had similar breed prevalence, with the majority French Bulldogs and Dachshunds, justifying the chondrodystrophic high incidence. In addition, the *t*-test revealed no statistically significant difference between groups in the DPP^−^ dogs.

### 4.1. DPP^+^ Dogs

The ambulation rate obtained in the SG was 99.4% (*n* = 167), with one dog achieving ambulatory paraparesis (but without the ability to perform 10 consecutive steps without falling). Between the SG and CG, there was a strong statistical significance regarding ambulation recovery (*p* < 0.001).

Within the SG, 68% of ambulation was achieved from T3 to T4 (Figure 6)—a similar result found in previous studies [65,97,98]. An ambulatory state was considered when OFS ≥ 11, and at T4, the OFS mean obtained in the DPP^+^ dogs was 11.0, higher than that previously reported by Zidan et al. (2018) [96] at the same time, which obtained an OFS mean of 7.87 in the basic group and 7.73 in the intensive group.

Furthermore, at time point T5 (day 30), a maximum result in the graphic curve was shown: 94.4% (156/167) of ambulation, achieving an OFS mean of 11.8. The following decline in the graphic curve may be explained by the remaining small number of dogs in the next time points, not allowing the results to be discussed.

In this study, all dogs from the SG underwent the same INR protocol, which differs from most protocols, given the early start of the underwater treadmill (UWTM) training (2–7 days after admission) in contrast to the 7–14 days reported by Zidan et al. (2018) [96]. Moreover, a major difference was the association of electrical stimulation with FES and not NMES, complementing the TESCS. Early training has been previously reported as having a major importance in recovery [6,99,100].

The BWS training was based on repetitive movements, depending on intensity, volume, and duration [6], with some clinical evidence suggesting that this training may improve the excitatory influence of descending pathways. In this study, all SG dogs performed locomotor training, in which the parameters were chosen according to different authors [45,53,54,55,56]. Slope training was also performed (inclination 10–25°) to promote the standing up ability [47,57].

This INR applied to DPP^+^ dogs was shown to be a feasible, safe, and repeatable protocol—able to be performed with some knowledge in the field of neuro-rehabilitation and acquiring a land treadmill and a UWTM.

Furthermore, electrical stimulation by FES may potentiate large-diameter motor neurons recruitment capable of fast conduction velocity fibers instead of recruiting small diameter motor neurons, which are slower and more susceptible to fatigue [101,102]. FES may also increase the muscle tonus in the hindlimbs and enhance the polysynaptic reflex, essential in this population of dogs that showed an absent or decreased withdrawal reflex [66,67,103,104].

This multidisciplinary treatment also included TESCS, which is considered a noninvasive and nonpainful neuro-modulation modality. TESCS has been proposed to induce stimulation through multi-segmental interactive and synergistic pathways, which combine the central components of motor descending paths and ascending sensorial paths [102], recruiting a diverse population of motor neurons by projecting sensory and intraspinal connections [75].

### 4.2. DPP^−^ Dogs

In regard to the assessment of DPP^−^ dogs, the OFS has limited value, given the fact that it is unable to evaluate the neurological signs based on peripheral reflexes—mainly, the evaluation of the flexion/extension locomotor pattern. Thus, there is a need to implement a precise and restricted outcome with the FNRS-DPP^−^ scale (published at the 31st ESVN-ECVN Symposium in Copenhagen 2018 [44]).

The total DPP^−^ dogs were 137, and 94 dogs were within the SG. The ambulation rate within this group was 55.8% (55/94), and when comparing the DPP recovery between the SG and CG, there was 33.2% (35/94) recovery in the SG, higher than the CG, with a tendency to be a significant statistical difference (*X^2^* (1, *n* = 137) = 3.597, *p* = 0.058), although a lower number, when compared to the results published by the CANSORT-SCI (2021) [105], refers to nearly 60% of the DPP recovery [105,106].

The DPP recovery assessment in the CG was very limited, because these were retrospective data, with most dogs euthanized by the owner´s decision after 3 weeks, a decision that did not allow for investigating the theory that permanent DPP^−^ dogs may demonstrate spontaneous motor recovery over time [2,107,108,109].

This reflects the major importance of 22 dogs in the SG that regained ambulation by SRL—37.2% in a maximum period of 3 months. This percentage was higher than the one reported by Olby et al. (2003) [107], with 32% (7/18) that regained ambulation within 9 months on average (range 4–18 months). Furthermore, among a cohort of 94 dogs examined in a chronic setting, nine became ambulatory within a median time of 12 months (range 3–89 months) [109].

On the other hand, Gallucci et al. (2017) [108] found, via a retrospective review, a median time of only 75 days to regain ambulation by SRL (range, 16–350 days), with 59% (48/81) of the DPP^−^ dogs recovering and a full-time hospitalization regime implemented 81% of the time; however, this had mixed etiology, including acute IVDE or trauma (e.g., vertebral luxation/fracture).

Most studies reported that a shorter average time to achieve ambulation may be related to an early post-injury intensive rehabilitation, which could have a positive influence on recovery [2,108]. In these studies [108,109,110], participants had T3–L3 as the major neuro-location, which is similar to the present study.

In this type of population, body weight distribution allows body support in the forelimbs, with the ability to postural stand but not regain ambulation, which could be explained due to the larger demand on the supraspinal postural control that is needed to maintain the balance that may be missing after severe injury [2].

Several investigations have reported that multimodal approaches may facilitate motor recovery and are useful in improving the outcome, in combination with additional approaches directed to the lesion epicenter [2,111,112,113,114].

To Gallucci et al. (2017) [108], spinal walking is a “reflex gait with complex dynamic interactions between CPG of pelvic limbs and proprioceptive feedback from the body in the absence of superior control by the brain after complete spinal cord damage”. For the present study, it is the author’s belief that the future aim for INR techniques is to develop strategies that directly target the spinal cord injury and limit the secondary injury, enhancing axons regeneration and/or increasing the compensatory plasticity of the persisting tissue. Thus, it is essential to understand broad knowledge regarding injury and recovery mechanisms in order to develop new strategies.

The INR protocol focused on the ability to perform the flexion/extension locomotor pattern, settled around the specific needs of each dog, and promoted exercises based on flexor reflex and crossed extensor reflex stimulation. The UWTM locomotor training was critical for the DPP^−^ dogs, facilitating the possibility to regain the flexion/extension pattern [108]. In Martins 2021 [104], randomization in different locomotor training groups was only possible after 15 days of UWTM training, warranting that possible spinal chock dogs were stimulated at a maximum level with a multidisciplinary treatment.

As a part of the treatment protocol, some authors recommend cage rest for 6–8 weeks [106,115], preventing future traumatic injury and decreasing pain and inflammation [116,117]. It is known that INR may be beneficial to locomotor recovery in a post-surgical patient, mainly in populations similar to our SG, which are severely affected dogs. Studies have shown that intensive locomotor training can promote anatomic and physiologic changes, allowing an improved motor function [118,119].

This study is in accordance with Zidan et al. (2018) and Jeong et al. (2019) [96,120]; however, the latter received decompressive surgery alone, reporting only 17%, which is much lower than the typical results reported in the literature, which usually range from 50 to 60%.

Moore et al. (2020) [106] refer to a similar training protocol when compared to our own, stating that locomotor training (land or UWTM) may help to regain movement. Thus, these types of rehabilitation strategies should be rationally implemented and initially encouraged in the recovery process for IVDE [121], as already shown in human medicine [122].

SG ambulation was achieved in 58.4% (55/94) of patients, in contrast to the 32.6% of the CG, probably because there is no case of SRL in the CG group. Thus, comparison between groups, regarding ambulation, has shown a strong statistically significant difference (*X^2^* (1, *n* = 137) = 7.311; *p* = 0.007).

In the SG, until T5, there was clinical evidence of PM, with dogs being euthanized in T3 and T4. The PM cases were 14.9% in total, a result that agrees with the previously reported studies [97,107,123,124,125,126]. However, the two dogs that remained until T5 had only signs of descending myelomalacia.

Jeffery et al. (2020) [93] suggested that an increase in the ambulation rate and decrease in cases of PM could be associated with the durotomy surgical technique as a means to examine the spinal cord and potentiate tissue perfusion, with an approximately 4% reduction.

In the present study, regarding DPP recovery, the greatest percentage occurred after T5 at 74.3% (26/35), as well as the appearance of the SRL, which was only present in T7 and T9. Both situations occurred after the pharmacological support with 4-AP. Thus, it is the author’s belief that the combination of these neurorehabilitation modalities, locomotor training, and 4-AP administration may help in recovery within a maximum of 3 months, including the possibility of a spontaneous plasticity contribution.

4-AP is a potassium channel antagonist that may improve hindlimb motor function in chronic thoracolumbar spinal cord dogs [81,114,127]. This beneficial effect could be achieved through the enhancement of central conduction by anatomically intact axons traversing the injury site, as well as direct synaptic effects [29,128,129].

Dogs that did not regain DPP throughout the 3 months of hospitalization were monitored by the FNSR-DPP^−^ scale in each time point outcome. The appearance of the flexion/extension locomotor pattern was a critical outcome, which was the moment of starting 4-AP administration. In Figure 9, the graphic represents a gradual increase throughout the time points with an ascending curve, although not constant between each time point. Substantial evidence at the beginning of the curve may be explained by the unlocked inhibition of the nervous system pathways, followed by a fast excitation.

In T7 and T8, there is a slight decrease in the graphic curve, given the fact that 15 dogs were clinically discharged within this period. In T9, the mean scale punctuation was 9.1, although 22 dogs achieved ambulation by SRL. All 22 dogs reached a minimum FNRS-DPP^−^ score 14, and the mean of FNRS-DPP^−^ ≥ 14 was only achieved in the follow-ups F3 and F4.

This mean that the scale punctuation may be justified by each variability (ranging from dogs with PM with lower scores to dogs with higher scores) and by the fact that dogs were clinical discharged throughout the process, decreasing the number of dogs that remained and were in a worse neurological state.

For Olby et al. (2020) [130] and CANSORT-SCI (2021) [105], approximately 60% of dogs with IVDE recovered DPP and ambulation by 6 months after SCI. In the present study, we had 57.4% of ambulation by 3 months, which might have been influenced by the INR that helped to decrease the time of recovery. Moreover, in the same study, 31% of dogs did not achieve DPP but regained the ability to walk within a mean time of 9 months (range 2–28 months), similar to other authors [115]. In our study, we achieved a higher prevalence with 37.3% (22/59) but at a period of 3 months, possibly leading to a better understanding from the owners when discussing prognosis and decreasing future euthanasia. In regard to dogs that regained DPP (*n* = 35), all were monitored with the OFS scale at each time point. The maximum OFS mean was 12.3 at T0, always increasing and maintained in the 6-month and one-year follow-ups, proving the spinal cord memorization property in the neural regeneration process.

The study limitations included the inability to achieve a level of power (1 − β) of 0.99 and an α (type I error) of 0.05 for each group, a lack of a prospective control group with the same conditions and with the same study group size, and the need to statistically relate the presence of peripheral reflexes with ambulation recovery and a nonexistent biomarker of regeneration, usually associated with structural proteins (e.g., glial fibrillary acidic protein—GFAP), although this would imply highly sensitive measuring techniques.

## 5. Conclusions

In acute IVDE dogs classified with OFS1 or FNRS-DPP^−^ grade 0/1 and after surgical management, the outcomes obtained were difficult to attribute to spontaneous recovery or with intensive rehabilitation. However, the results suggested that, with INR ambulation, success could be improved, mainly regarding time, within an estimated period of 3 months. Thus, its implementation may be useful in acute post-surgical IVDE dogs.

## Figures and Tables

**Figure 1 animals-11-03034-f001:**
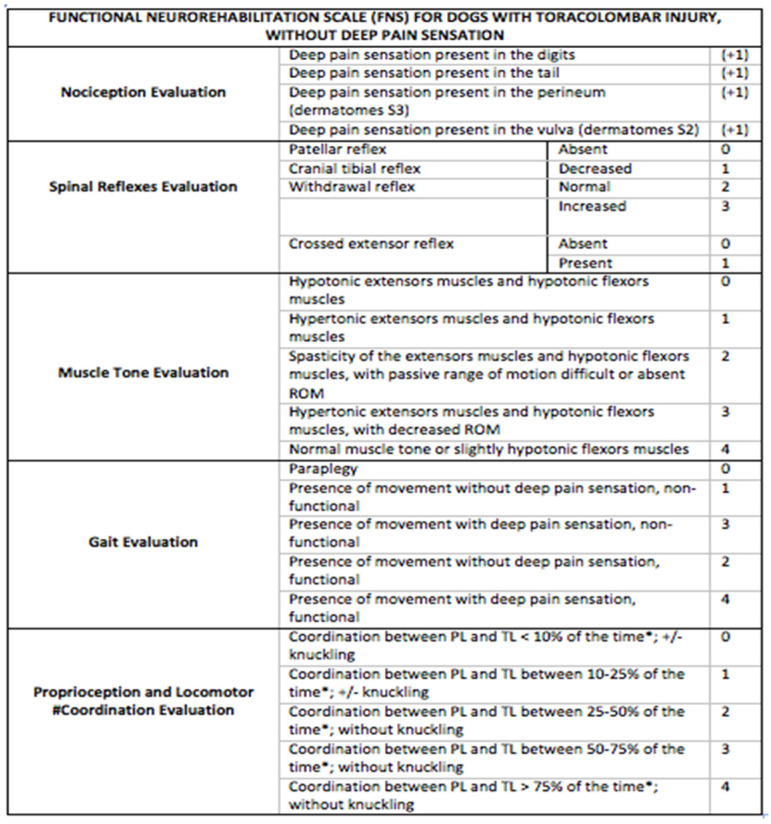
Functional neurorehabilitation scale for dogs with thoracolumbar spinal cord injury without deep pain perception. 31st Annual Symposium of the ESVN-ECVN [44].

**Figure 2 animals-11-03034-f002:**
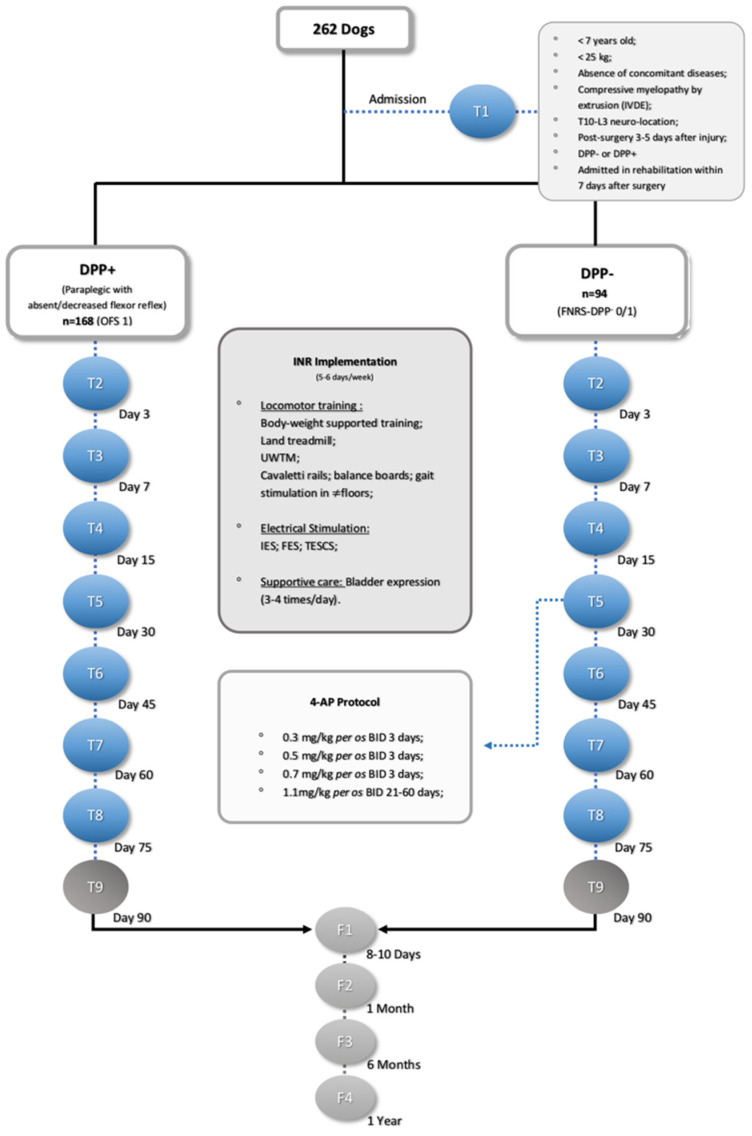
Flow diagram (as recommended by Strengthening the Reporting of Observational Studies in Epidemiology, STROBE, guidelines) illustrating the study design. IVDE: intervertebral disc extrusion; DPP: deep pain perception, INR: intensive neurorehabilitation, OFS: open field score, FNRS-DPP^−^: functional neurorehabilitation scale for thoracolumbar SCI dogs without DPP, UWTM: underwater treadmill, IES: interferential electrical stimulation, FES: functional electrical stimulation, TESCS: transcutaneous electrical spinal cord stimulation, and 4-AP: 4-aminopyridine.

**Figure 3 animals-11-03034-f003:**
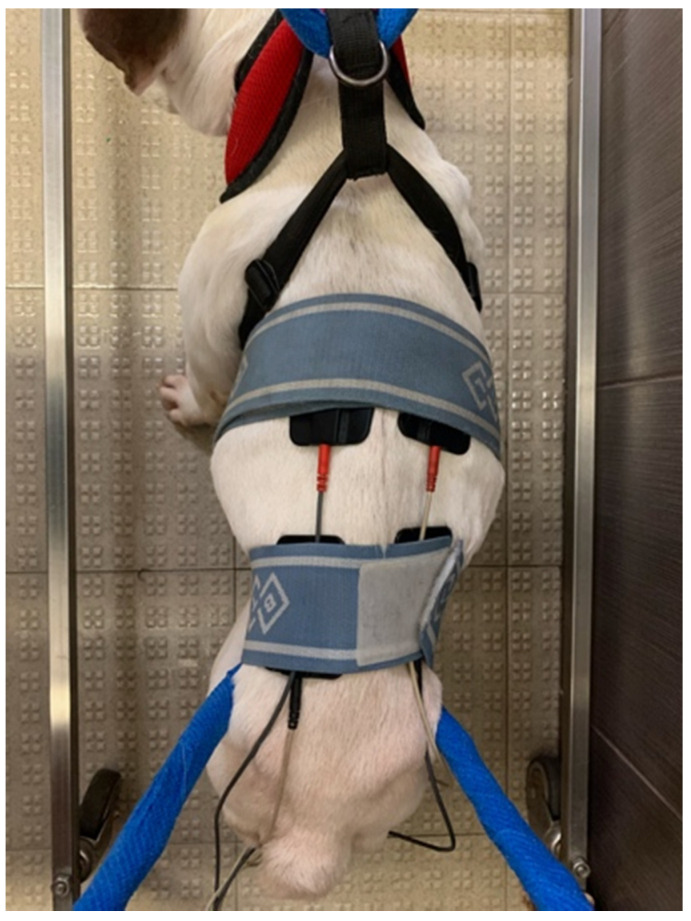
Interferential electrical stimulation. Four electrodes crossed at a 90° angle near the spinal hyperesthesia region.

**Figure 4 animals-11-03034-f004:**
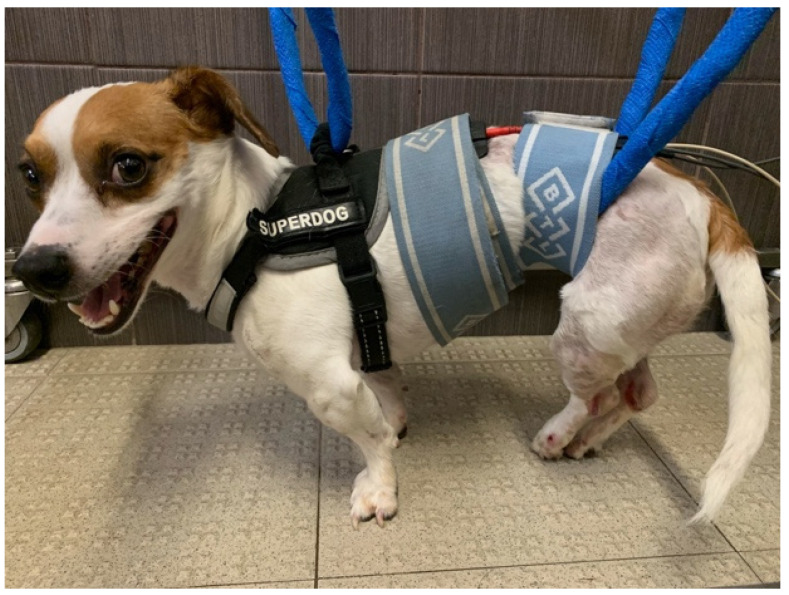
Transcutaneous electrical spinal cord stimulation. One electrode of each canal at the T11 to T12 region and the others at the L7–S1 region.

**Figure 5 animals-11-03034-f005:**
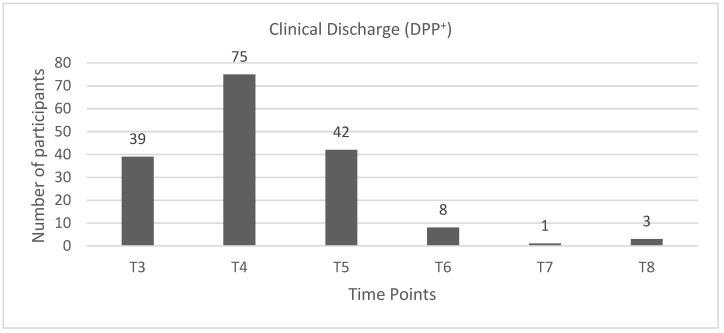
Graphic describing the number of DPP^+^ dogs in the study group (SG) with clinical discharge in each time point. DPP^+^: deep pain perception-positive, T3: day 7, T4: day 15, T5: day 30, T6: day 45, T7: day 60, and T8: day 75.

**Figure 6 animals-11-03034-f006:**
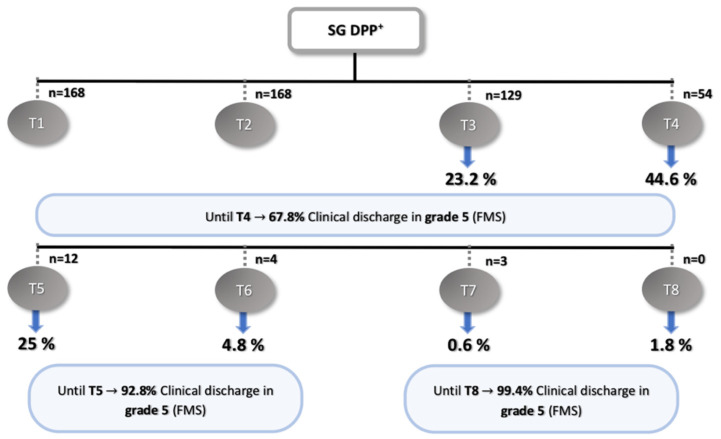
Flow diagram describing the results of the DPP^+^ dogs in the study group. SG: study group, DPP^+^: deep pain perception-positive, T1: admission, T2: day 3, T3: day 7, T4: day 15, T5: day 30, T6: day 45, T7: day 60, T8: day 75, and FMS: Frankel modified scale.

**Figure 7 animals-11-03034-f007:**
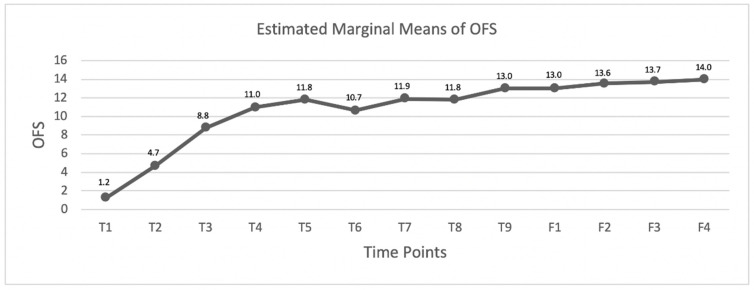
Graphic describing the OFS mean evolution of the DPP^+^ dogs in the study group (SG). T1: admission, T2: day 3, T3: day 7, T4: day 15, T5: day 30, T6: day 45, T7: day 60, T8: day 75, T9: day 90, F1: 8–10 days follow-up, F2: 1-month follow-up, F3: 6-months follow-up, and F4: one-year follow-up.

**Figure 8 animals-11-03034-f008:**
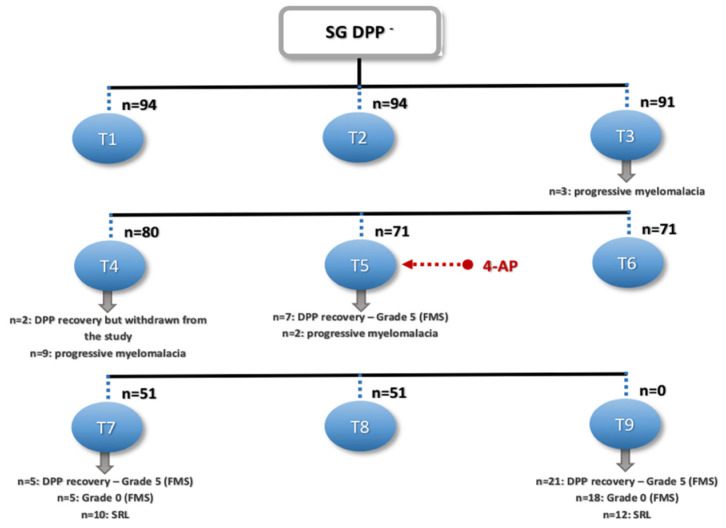
Flow diagram describing the results of the DPP^−^ dogs in the study group. SG: study group, DPP^−^: deep pain perception-negative, T1: admission, T2: day 3, T3: day 7, T4: day 15, T5: day 30, T6: day 45, T7: day 60, T8: day 75, T9: day 90, FMS: Frankel modified scale, and SRL: spinal reflex locomotion.

**Figure 9 animals-11-03034-f009:**
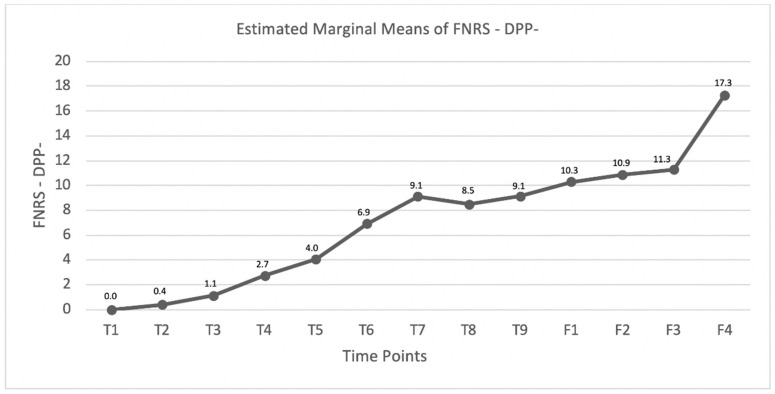
Graphic describing the FNRS-DPP^−^ mean evolution within the DPP^−^ dogs in the study group (SG). T1: admission, T2: day 3, T3: day 7, T4: day 15, T5: day 30, T6: day 45, T7: day 60, T8: day 75, T9: day 90, F1: 8–10 days follow-up, F2: 1-month follow-up, F3; 6-month follow-up, and F4; one-year follow-up.

**Figure 10 animals-11-03034-f010:**
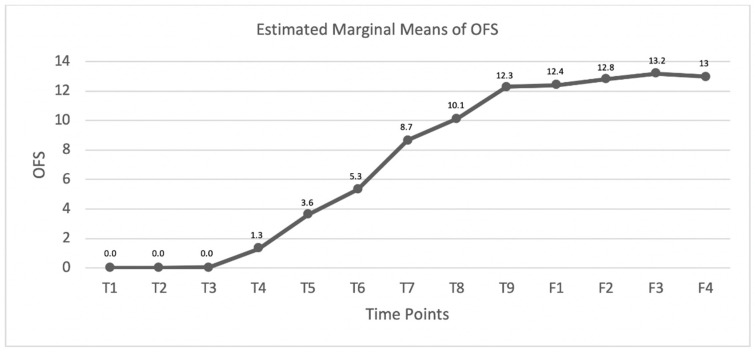
Graphic describing the OFS mean evolution within the DPP^−^ dogs that recovered DPP in the study group (SG). T1: admission, T2: day 3, T3: day 7, T4: day 15, T5: day 30, T6: day 45, T7: day 60, T8: day 75, T9: day 90, F1: 8–10 days follow-up, F2: 1-month follow-up, F3: 6-month follow-up, and F4: one-year follow-up.

**Table 1 animals-11-03034-t001:** DPP^+^ population characterizations at admission (*n* = 230).

	DPP^+^ (*n* = 230)			SG (*n* = 168)			CG (*n* = 62)		
	Mean (SD)	95% CI	Median	Mean (SD)	95% CI	Median	Mean (SD)	95% CI	Median
Age (years)	4.07 (1.574)	3.86–4.27	4.00	4.07 (1.642)	3.86–4.27	4.00	3.76 (1.339)	3.42–4.10	4.00
Bodyweight (kg)	8.79 (4.032)	8.27–9.32	8.00	8.79 (4.484)	8.27–9.32	8.00	7.79 (2.159)	7.24–8.34	7.00

Abbreviations: DPP: deep pain perception, SG: study group, CG: control group, CI: confidence interval, and SD: standard deviation.

**Table 2 animals-11-03034-t002:** DPP^−^ population characterization at admission (*n* = 137).

	DPP^−^ (*n* = 137)			SG (*n* = 94)			CG (*n* = 43)		
	Mean (SD)	95% CI	Median	Mean (SD)	95% CI	Median	Mean (SD)	95% CI	Median
Age (years)	4.03 (1.576)	3.76–4.30	4.00	3.90 (1.566)	3.58–4.23	4.00	4.30 (1.582)	3.82–4.79	4.00
Body weight (kg)	8.14 (3.218)	7.59–8.68	8.00	8.51 (3.466)	7.80–9.22	8.00	7.33 (2.437)	6.58–8.08	7.00

Abbreviations: DPP, deep pain perception; SG, study group; CG, control group; CI, confidence interval; and SD, standard deviation.

## Data Availability

The data presented in this study are available upon request from the corresponding author.

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
