# Peer review of "A Controlled Clinical Study of Intensive Neurorehabilitation in Post-Surgical Dogs with Severe Acute Intervertebral Disc Extrusion"

_animals, 2021, doi:10.3390/ani11113034_

Round 1
Reviewer 1 Report
The authors showed that the intensive neurorehabilitation, as part of a multimodal training protocol could improve ambulation in acute post-surgical intervertebral disc extrusion dogs. It is a very interesting study, well explained, with sections presented in a balanced and coherent way. Introduction is properly documented, and the data are properly discussed.
I suggest minor revisions providing some comments:
lines 37 and 692: I believe that the correct word is "mainly" rather than manly.
line 171: please report what nonsteroidal anti-inflammatory or corticosteroid drugs were used.
Reviewer 2 Report
The present manuscript is of high clinical relevance and involved a large number of animals, considering the case selection. However, there are several formal aspects that diminish the quality of the manuscript and that should be addressed before publication. The following comments are not extensive.
According to the Instructions for Authors, the Simple Summary must not include abbreviations (DPP, for example) and thus must be altered.
The English language needs extensive revision by a native speaker. There are many style and grammar issues. For example, in line 35 "didn't recovered" instead of "didn't recover" or "failed to recover"; in 37 "manly" instead of "mainly" (repeated in line 692); in line 45 do the authors mean "thus" or "although"?; the phrase "The same..." in line 74 should be re-written. In line 114 "All participants had less than 7 years old, less than 25 kg" would probably be more correct as "All animals in the study were less than 7 years old and weighed less than 25kg". The style and grammar issues hinder the clarity of the text.
In line 98 it would be important to clarify whether the imagiology findings were supported by the neurological examination findings.
The authors should clarify if in line 149 they do want to refer to "spinal cord palpation" or instead mean spinal palpation or palpation of the spine.
Phrases such as the ones beginning in line 222 and 388 should probably be re-written for improved clarity and correction.
In line 170 and 171 it is said that all animals were treated with non-steroidal or corticosteroids. However, this information, possibly relevant for recovery is not detailed in terms of which drugs were used, doses, to animals in either group, and whether it was investigated if the different medication impacted the outcome. Albeit it is understandable that protocols might have changed throughout the period of the study, more detailed information is desirable, possibly condensed in a table.
The authors chose a parametric test, an independent samples t test. Did the authors verify normal distribution and equality of variances?
The control and experimental groups had different sizes. Did the authors consider this when choosing the statistical analysis?
Reviewer 3 Report
Manuscript writing is confusing.Author Response
Regarding the general comments, we thank the reviewer for all of the comments and suggestions. We re-wrote the manuscript, also considering the suggestions of other reviewers and hope we could be clearer in our writing. Also the English was revised by a native person and the MDPI authors service.
We think with all suggestions that were made, our manuscript was improved and became a better version.
Thank you once again.
Round 2
Reviewer 2 Report
In the revised and much improved document, there are issues still to address that I think were not answered in the document or in the author's reply, namely:
- use of AINEs and steroids: is not detailed which drugs were used the to animals in either group, and whether it was investigated if the different medication impacted the outcome. Protocols may have changed throughout the period of the study, more detailed information is desirable, possibly condensed in a table.
- The authors chose a parametric test, an independent samples t test. Did the authors verify normal distribution and equality of variances? The control and experimental groups had different sizes. Did the authors consider this when choosing the statistical analysis?
Round 3
Reviewer 2 Report
Including in the text which test was used for homogeneity of variances (in this case, Levene's, but there are others, less used such as Barlett's and Cochran’s ) and expressively mentioning this was done, along with normality distribution tests is relevant and part of sound Methods description, even if SPSS was used.
The same applies to the discussion of the effects of group unequal size, considering the test chosen to compare the two groups. Since unequal-sized groups may threaten the validation of the results, this should also be mentioned and discussed, again even if SPSS was used.
https://libguides.library.kent.edu/spss/independentttest
Excellent clinical work deserves scientifically sound communication.